# Hybrid Power System Optimization in Mission-Critical Communication

**Sonia Leva** [1,*] **, Francesco Grimaccia** [1] **, Marco Rozzi** [1] **and Matteo Mascherpa** [2]

[1]  Department of Energy, Politecnico di Milano, 20156 Milano, Italy; francesco.grimaccia@polimi.it (F.G.); marco.rozzi@mail.polimi.it (M.R.)
[2]  GEG Srl, 24020 Cene (BG), Italy; matteo.mascherpa@gegsrl.com
*   Correspondence: sonia.leva@polimi.it; Tel.: +39-02-23993709

**Abstract:** One of the common problems faced by Telecommunication (TLC) companies is the lack of power supply, usually for those appliances with scarce chances of grid connection often placed in remote zones. This issue is more and more critical if the radio network has the specific task of guaranteeing the so-called "mission-critical communications". This manuscript aims to propose and assess a viable solution to optimize the power supply and maintenance operations required to assure the proper functionality in such critical and remote sites. In particular, the main goals are defining a method to select the critical sites in an extensive and composite radio system and designing the hybrid power system in a way to improve the service availability and technical-economic benefits of the whole mission-critical TLC system. Finally, the proposed method and related procedures are tested and validated in a real scenario.

**Keywords:** mission-critical application; hybrid power system optimization; genetic algorithms; economics in PV-Hybrid system

---

## 1. Introduction

The lack of stable and reliable electric grid connections in remote sites is quite a common issue in many telecommunication systems to provide proper radio coverage features. Such a problem is quite common in mission-critical scenario applications, especially in situations where the communication infrastructure has to guarantee a continuous standby solution to ensure effective support to emergency service operators in particular "Mission-Critical" (MC) scenarios.

In order to overcome such a critical point and ensure a steady power source for MC network nodes, a viable solution is the adoption of Hybrid Power Systems (HPS). Generally, Internal Combustion Engines (ICEs) fuelled by diesel have been used as a standard technical solution for this purpose due to properties such as high flexibility and reliability. However, diesel-generated power is usually more expensive than the average cost of grid electricity. Lately, driven by a power generation economic saving due to the contribution of Renewable Energy Sources (RES), the adoption of hybrid renewable power systems (e.g., photovoltaic modules, wind turbines) has been proposed in the literature [1]. These widely applied solutions [2–4] are usually combined with a battery energy storage system and fuel cell, to face the intrinsic intermittency of wind and solar generators [5,6].

The design of the HPS is performed by using commercial software. This analyses system performance from optimization and economic aspects [2,3], but also by using tools developed ad hoc for the analysis of particular cases [6,7] in a way to take into account the peculiarities of the application but also forecasting RES power production [8,9] and load consumption [10].

Among optimization algorithms proposed in the literature, heuristic optimization methods [11] are the most common due to their ability to achieve the near-optimal value for the fitness function.

Some of the heuristic algorithms used are: Genetic Algorithm (GA), Simulated Annealing algorithm (SA), and Particle Swarm Optimization (PSO). Some heuristic optimization methods for RES based on lithium-ion are presented in [12]. In particular, [13] presents a constrained multi-objective optimization framework for battery charging management solved by using Non-dominated Sorting Genetic Algorithm II (NSGA-II).

The scope of this research is to develop and assess a method to minimize the number of interventions required to restore the correct functioning, increasing the power supply availability and bringing both technical and economic benefits to the power supply of some peculiar telecommunications sites part of complete and complex TLM systems for critical communication. As already mentioned in [14], this goal can be reached by using also fuel cell storage systems based on hydrogen in strategic sites for the telecommunication system under analysis, due to the economic impact of Proton Exchange Membrane Fuel Cell (PEMFC) technology.

The first step is to select critical and strategic sites in a wider and more complex radio network, where it is potentially profitable to install a complete and complex HPS. The choice of these strategic sites is linked both to technical questions (relevance in the TLC system), but also geographical issues connected either to the availability of primary sources (such as irradiance and wind) and to the reachability of the site. In this paper, a detailed and completed procedure that defines a relevant number of adapted criteria able to properly characterize the specific site and performs a criticality analysis with the aim of identifying the proper solutions which require investments to ensure power supply is presented and discussed. Furthermore, a specifically tailored tool based on Genetic Algorithms to optimize the hybrid power system sizing in terms of high levels of RES penetration for energy independence of specific sites in the critical mission telecommunication system is developed. The tool returns the characteristics and the number of components necessary to ensure the energy independence of the selected site in a way to increase the target of improving service availability, extended lifetime and technical–economic benefits of the TLC system.

The main contribution of this manuscript is the development of a proper methodology and ad hoc selection criteria to define radio network sites that can assure energy supply independence to maximize system resilience under natural disasters or critical events. These criteria take into account both technical and economic aspects since the results of the analysis are applied to a real case study, namely a regional and multi-tenancy complex network in the north of Italy. The Professional Mobile Radio (PMR) is a niche market considering just a regional or even a national context, but public safety and communication infrastructures nowadays are globally representing strategic asset and related energy savings and sustainability issues are attracting more attention than in the past. Thus, the integration of renewable applications in this sector is essential and not sufficiently considered in the state of the art of current systems. Furthermore, with the aim to validate and make off-grid HPS with high penetration of renewable sources really applicable in the field, not only the environmental issues but also the financial benefits should be taken into account with respect to typical high investment costs of such mission-critical systems.

This paper is structured as follows: Section 2 introduces a methodology for assets devoted to critical communication services and defines criteria to rank telecommunication sites in a complex network in terms of criticality index. Section 3 describes a complete tool to optimize the sizing of hybrid power systems for supply network sites. Section 4 reports a real case study and the results obtained by numerical simulation. In Section 5, some conclusions also based on stress analysis are reported.

## 2. Criticality Analysis of Mission-Critical Asset

MC communications always need technical solutions characterized by peculiar properties such as availability, robustness and security that are essential to ensure reliable services. MC typical end-users are public safety, firefighters, emergency medical services, military users, critical infrastructure operators, utilities, etc.

Today it is essential to manage the risks of failure of such systems, performing a criticality analysis based on a structured method, which follows the goals of the end-users organization and the impact that their inadequate operations might lead.

In this section, we will define the main criteria to perform system optimization, specifying different site characteristics, failure modes, operational and maintenance activities to properly define a criticality index for each network site.

### 2.1. Criteria Definition and Parameter Calculation

The type of the strategic level and criticality of a specific site is mainly technical, with related economic constraints, and they can be divided into three different categories: site characteristics, malfunctioning causes, operational and maintenance activities, here listed as follows with the definition of related indexes.

#### 2.1.1. Site Characteristics

Number of users (0–1): for every site in the specific network the weighted index is computed by the ratio of the number of hosted users ($N_{users}$) and the maximum number of clients of the radio sites as follow:

$$I_{users,i} = \frac{N_{users,\,i}}{MAX[N_{users,i}]} \tag{1}$$

Site typology (0–1): a weight is assigned as a function of connection among sites in TLC network; as a matter of fact, a terminal site (=0) is less important than a cascade or series site (=0.5), instead a nodal site is the most important one (=1).

Radio covered area (0–1): the index:

$$I_{covered\ area,i} = \frac{A_{covered,i}}{MAX\big[A_{covered,i}\big]} \tag{2}$$

is computed based on the coverage study ($A_{covered}$), performed by specific software and taking as denominator covered area in the specific network.

Radio traffic criticality (0–1): it is equal to the Erlang value [15] linked to the time during the radio site is in transmission mode (TX).

Accessibility and reachability (0–1): it depends on the type of vehicle that needs to reach the site in the critical seasons: standard vehicle = 0; off-road vehicle = 0.5; helicopter or others = 1.

#### 2.1.2. Malfunctioning Causes

Meteorological criticality (0–2): for each site, the meteorological data are analysed basing on-site measured data or considering as reference the closest meteo-station is considered. In each site the number of hours during which the quantity of cumulated fallen rain, snow and wind speed values which exceed the predefined range are calculated and according to these results a risk index is attributed at the site. The thresholds are defined as follow:

- Cumulated fallen rain in 24 h: intense: 30–50 mm, intense and persistent: 50–80 mm, large quantity: >80 mm;
- Quantity of fallen snow in 24 h: intense: 40–60 mm, abundant: 60–80 mm, large quantity: >80 mm;
- Wind speed: strong gusts: 75–90 km/h, stormy: 90–120 km/h, cyclonic: >120 km/h.

The value of this index could be 0, 1 or 2 corresponding to *ordinary-*, *moderate-* or *elevate-risk* respectively.

Lack of power supply (0–2): for each site, the occurrence of registered power shortage and their time-duration is taken into account. Indeed, a lack of energy supply can be easily managed by the storage system present in the site, while a longer usually requires prompt technical intervention.

$$I_{NLPS} = \frac{N_i}{MAX[N_i]} + \frac{\Delta t_i}{MAX[\Delta t_i]} \tag{3}$$

### 2.1.3. Operational and Maintenance Activities

Number of operations on-site (0–1): the number of interventions per year in a site is evaluated. On the base of experience, the following thresholds are fixed: good performance site (from 0 to 4 maintenance interventions) = 0; normal performance site (from 5 to 8 maintenance interventions) = 0.5; critical performance site (maintenance interventions > 8) = 1.

### 2.2. Criticality Index Estimation and Site Ranking

Balancing the indexes on the maximum number of users, maximum covered area and maximum number of registered power outages and related time span allows keeping a reference value that is representative of the cluster of considered sites, to determine which are the worst and the best site of a specific network.

The final result of the criticality analysis is an overall Criticality Index (CI) with a value ranging between 0 and 10, as reported in Table 1.

**Table 1.** Criticality Index rank and description.

| CI | Description |
|---|---|
| 0–3 | Low critical site. It is characterized by a high availability, high reliability, low meteorological risk and it is not so strategic in the network |
| 4–6 | Medium critical site. Two mains different cases can apply. A strategic site in terms of the TLC network, but a low-risk site from the point of view of maintenance operations. A site with a low value of availability and reliability, but not so strategic in terms of telecommunication system (e.g., terminal site). |
| 6–10 | High critical site. It is a strategic site, but with poor reliability and availability. Economic investments, with the aim of increasing the continuity of service, are justified. |

## 3. Hybrid Renewable Power Systems Models

The hybrid power system analyzed in this paper is depicted in Figure 1. It includes as renewable generators a PhotoVoltaic (PV) system and a Wind Turbine (WT) included their DC/DC converters; as storage a bank of Lithium-ion batteries (BT) and a Proton Exchange Membrane Fuel Cell (PEMFC) systems; a diesel generator (DG) as back-up system; a DC/AC converter (CONV); and the Radio Site (RS) as the main load. The HPS design is a challenging task that has been deeply studied in the literature [1,6].

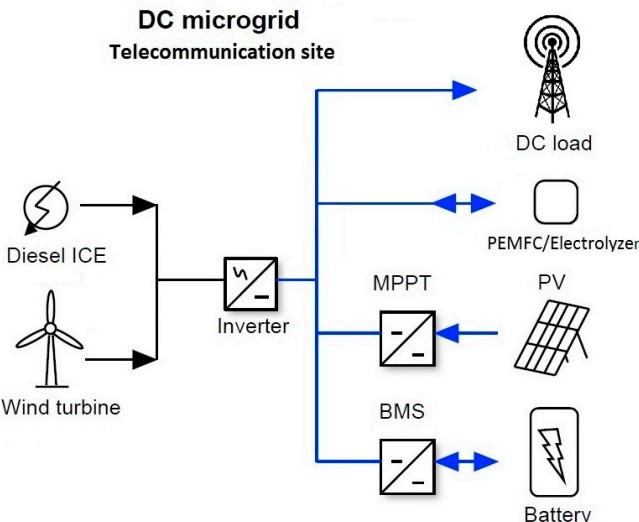

**Figure 1.** Hybrid Power Systems (HPS) for Mission-Critical (MC) telecommunications site.

The HPS design is a challenging task that has been deeply studied in the literature. In this paper, to represent the main peculiarities of MC communications systems, specific software is designed to optimize and manage the HPS for the MC site considering a time resolution equal to 1 h. Three main sub-algorithms compose the algorithm, depicted in Figure 2: the Meteorological Module, the Optimization Module and the Simulation Module.

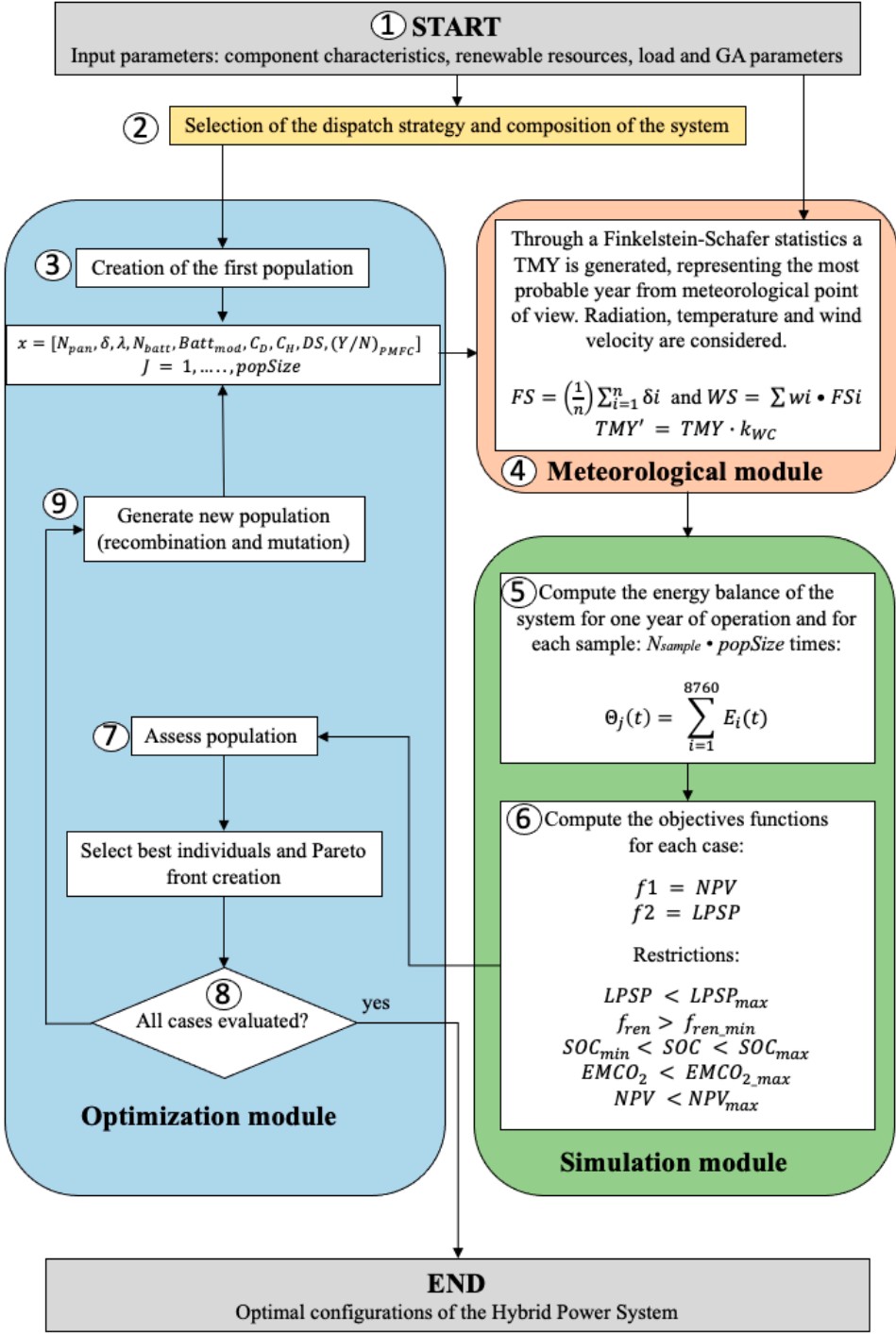

**Figure 2.** Block diagram of the proposed algorithm.

The economic and technical characteristics of the HPS components, main features of the renewable resources and supplied loads, and the structure of the GA are the input parameters set in step ①.

The end-user defines the dispatch strategy and the structure of the system is selected in step ②. In step ③ the first population of the GA is randomly generated by the individuals (*popSize*).

Step ④ is dedicated to the evaluation of the meteorological quantities. There are different possibilities to define temperature, wind speed and direction, irradiance, etc. [15–17]; in the present tool, the Typical Meteorological Year (TMY) is defined using the Finkelstein–Schaffer (FS) statistic [18].

In step ⑤, each solution is simulated *Nsamp* times under the uncertainty scenarios generated in step ④. The algorithm performs a total of $Nsamp \times popSize$ simulations of the HPS in each generation of the GA. Therefore, for each solution in the population, a matrix of energy balance results $\Theta_j(t)$ is obtained to perform system evaluation in the next step ⑥. In step ⑦, the algorithm checks for each individual the physical and economic constraints defined by the user, resulting in ranked solutions depending on the objective functions and the number of constraints out of range. In step ⑧, if the value of the number of samples is below the maximum, the population is updated by means of recombination and mutation operators (step ⑨). Candidate solutions are replaced with new individuals who should outperform the previous ones. After a complete run of population updates, the individuals are used in the simulation module to evaluate the objective function score, related limits and back to the optimization module again. This loop is repeated until the maximum number of examples are evaluated.

### 3.1. Optimization Module

The Optimization Module, depicted in Figure 3, is devoted to explore and find the best layout of HPS among the overall feasible configurations, solving the optimization problem as defined in the following equations:

$$\begin{aligned}
&\text{Minimizing}: \ obj_m(\overline{x}) \ m = 1, \ 2, \ \ldots., \ M \\
&\text{subject to}: \ cons_j(\overline{x}) \geq 0 \ j = 1, \ 2, \ \ldots, \ J \\
&\text{and} \ x_n^{(L)} \leq x_n \leq x_n^{(U)} \ i = 1, \ 2, \ \ldots, \ n
\end{aligned} \tag{4}$$

where:

- Variables vector, $x$. The considered solution corresponds to the number and type of components of the HPS, namely number PV modules ($N_{pan}$), tilt ($\Delta$) and azimuth ($\lambda$) angle of PV system, number of batteries ($N_{batt}$) and battery model ($Batt_{mod}$), diesel ($C_D$) and hydrogen ($C_H$) tank capacity, dispatch strategy ($DS$) and presence ($Y$) or absence ($N$) of the PEMFC:

$$x = \left[ N_{pan}, \ \delta, \ \lambda, \ N_{batt}, \ Batt_{mod}, \ C_D, \ C_H, DS, (Y/N)_{PMFC} \right] \tag{5}$$

- Objective functions. The HPS optimization takes into account two objective functions, namely a cost function and a power supply availability. The first Net Present Cost (NPC) is defined according to the equation:

$$NPC = IC + \sum_{i=1}^{Lifetime} NET_{CFactualized} \tag{6}$$

where *IC* (EUR) is the investment cost and the $NET_{CFactualized}$ (EUR) is the value of the net present cost actualized at the present. It is computed as follows:

$$NET_{CFactualized} = CF_{out,actualized} - CF_{in,actualized} \tag{7}$$

and the costs (*CF*) are actualized considering the nominal interest rate ($k_d{}'$), the inflation rate ($f$) and the lifetime of the project ($L_p$):

$$CF_{actualized} = CF \cdot \frac{k_d \cdot (1 - k_d)^{L_p}}{(1 + k_d)^{L_p} - 1} \tag{8}$$

and the annual interest rate ($k_d$) is:

$$k_d = \frac{k'_d - f}{1 + f} \qquad (9)$$

The second objective is related to the reliability of the system, and is computed through the Loss of Power Supply Probability (*LPSP*) shown in the equation:

$$LPSP = \frac{\sum_{i=1}^{8760} LPS(t)}{\sum_{i=1}^{8760} P_{load}(t) \cdot \Delta t} \qquad (10)$$

where *LPS* is the Loss of Power Supply (kW) and the *P* is the power consumption (kW) in the $\Delta t$ time period.

- The constraints functions are the limits related to the upper and lower boundaries of the design variables choice and the ones listed in Figure 3. Setting these limits, the algorithm searches automatically the feasible solutions in a limited hyperspace of variables, giving back only plants with a layout that respects these constraints.

- Optimization algorithm. The HPS sizing problem is solved using a multi-objective optimization procedure based on GA, i.e., the Non-dominated Sorting Genetic Algorithm II (NSGA-II) [19]. The NSGA-II has proven good behavior in solving complex problems due to the use of three main operators:

    a.  elitism, to avoid the loss of good solutions once they are found;
    b.  fast non-dominated sorting, a selection mechanism that ranks the individuals at different levels based on Pareto domain;
    c.  crowding distance assignment, a mechanism that exploits population diversity.

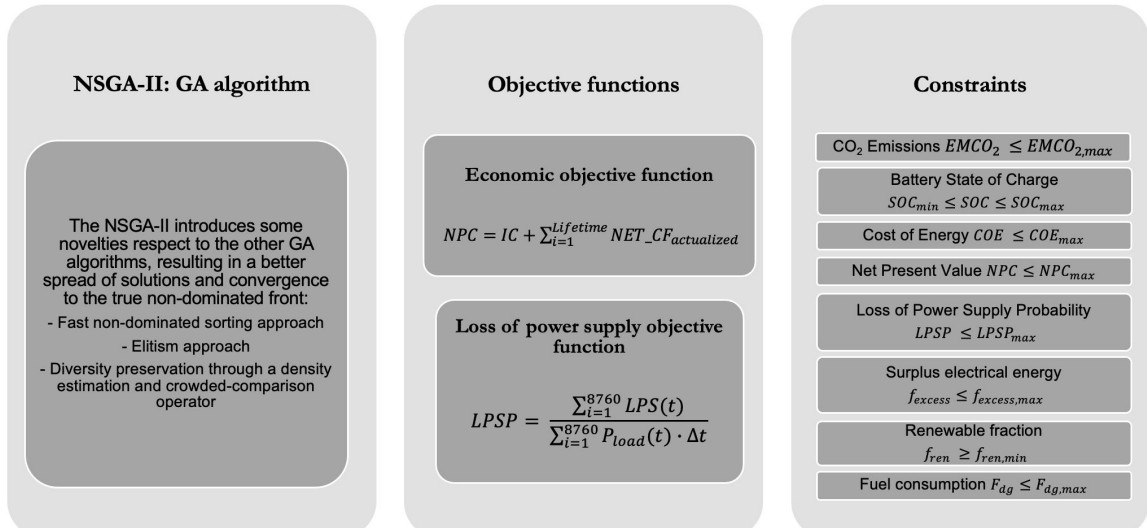

**Figure 3.** HPS model and sizing tool: Optimization module—Non-dominated Sorting Genetic Algorithm II (NSGA-II), objective functions and constraints.

### 3.2. Meteorological Module

A TMY is a set of meteorological data with hourly values in a year for a specific geographic area. The data are selected from measurements in a longer time period, normally 10 years or more, to have a good temporal variation. For each month in the year, the selection occurs from the year that is considered the most "typical" for that month. We started from minute by minute data to obtain such hourly data, and this approach allows filtering the inputs in the time domain.

TMY data sets are widely used in building design applications and in other technical fields including RES modelling.

The procedure should consider three steps.

For each month of the calendar year, five candidate months with Cumulative Distribution Functions (CDFs) for the daily indices that are closest to the long-term CDFs are selected. Candidate month CDFs are compared to the long-term CDFs by using Finkelstein–Schafer (*FS*) statistics.

$$FS = \frac{1}{n} \sum_{i=1}^{n} \left| CDF_m(x_i) - CDF_{y,m}(x_i) \right|$$

According to the Finkelstein–Schafer statistic, if a number $n$ of observations of a weather index $x$ are available and are sorted into an increasing order $x_1$, $x_2$, ..., $x_n$, the cumulative distribution function for the weather index is determined by a monotonic increasing function $CDF(x)$ as follows:

$$CDF(x) = \begin{cases} 0 & for\ x < x_1 \\ \frac{(i-0.5)}{n} & for\ x_i \leq x \leq x_{i+1} \\ 1 & for\ x \geq x_n \end{cases}$$

Because some of the indices can be more important than others, a weighted sum of the *FS* statistics is used to select the five candidate months that have the lowest index.

Then the five candidate months are ranked with respect to the proximity of the month to the long-term average and median.

Finally, the 12 selected months are lined up to obtain a complete year and discontinuities at the month are smoothed for 6 h each side using curve-fitting techniques.

### 3.3. Simulation Module

The Simulation Module output is a set of performance indices related to economic, reliability and environmental issues. These indices are considered by the Optimization Module to target the exploration phase in order to find the best HPS layout, taking into account meteorological values such as wind speed, solar irradiance, ambient temperature and load demand.

The algorithm is based on the hourly ($\Delta t = 1$ h) simulation of a full year of operation of the HPS for long-term simulations. The purpose of such a Simulation Module is to evaluate the objective functions and performance indices for each HPS configuration for a running time of one year, managing the interaction between different sources and loads in the system.

The models of the different components are briefly described referring to the reference list as follows, being common models, with the exception of the ones describing the battery bank and the shelter thermal behaviors. These last will be described more in-depth at the end of this section.

Photovoltaic (PV) systems. The mathematical model of the PV power generation system is based on the single point efficiency modeling of a PV module [20]. The temperature of the photovoltaic cell under operating conditions ($T_c$) is computed using the modified MRSSI correlation [21]. A power reduction coefficient takes into account all the losses not dependent on temperature (e.g., mismatching, shadow, dirty, etc.). The irradiance is estimated from the global horizontal irradiance [22] included in the meteorological dataset, as reported in [23].

Wind turbine (WT). The power generated by a WT ($P_{wt}$) can be approximated by its power curve, which indicates the power supplied by the WT as a function of the available wind speed, using a spline interpolation [24]. The real power generated by the WT is obtained according to the WT hub height and the air density modification due to the installation site altitude [25].

Diesel generator. The diesel generator [26] is a backup power source used as an emergency power-supply when the RSs are not available, and the battery bank is at its lowest State Of Charge (SOC).

The generator power output is modelled using the following linear function:

$$F_{dg} = F_0 \cdot P_{dgr} + F_1 \cdot P_{dg}$$

where $F_{dg}$ (L) is the instantaneous consumption, $P_{dgr}$ (kW) is the rated power, $P_{dg}$ (kW) is the output power, $F_0$ (L/h/kWrated) is the fuel intercept coefficient and $F_1$ (L/h/kW) is the slope coefficient.

Converter model. The model is capable of representing the converter efficiency over the full range of operating conditions [16,27]. The efficiency of the converter ($\eta_{converter}$) is:

$$\eta_{converter} = \frac{p - p_0 - k \cdot p^2}{p}$$

where p is the load fraction, and $p_0$ and k are specific to the inverter selected.

Battery Bank model. The mathematical model used to represent the battery integrates a performance model, a thermal model, a degradation model and a lifetime model. The performance model defines the amount of energy that can be extracted from the battery bank for each time step [28,29].

The SOC of the battery at each time is:

$$SOC(t+1) = SOC(t)\left(1 - \frac{\sigma \cdot \Delta t}{24}\right) + \frac{I_{bat}(t) \cdot \Delta t \cdot \eta_{ch-dch}}{C'_{bat}(t)} \tag{11}$$

where $\sigma$ (%) is the self-discharge coefficient, $I_{bat}$ (A) is the in–out current, $\eta_{ch-dch}$ is the charge and discharge efficiency and $C'_{bat}$ (Ah) is the battery modified capacity according to the temperature of the device.

The thermal model of the battery is sketched in different parts, to simplify the thermal analysis and the calculation of heat flows [30]. The energy balance is:

$$c_p m \frac{\left(T_{cell,internal,final} - T_{cell,internal,initial}\right)}{\Delta t} = Q_{heat} - \frac{kA}{d}\left(\overline{T_{cell,internal}} - T_{cell,surface}\right) \tag{12}$$

where $c_p$ (J/kg K) is the heat capacity of the battery computed as a weighted average of the overall capacity, $m$ (kg) is the mass of the battery internal, $k$ (W/m K) is the thermal conductivity, $A$ (m$^2$) is the cross-section area of the cell and $d$ (m) is the thickness of the cell. The $T_{cell,surface}$ can be computed in the function of $T_{amb}$ considering that the heat flow absorbed by the conduction effect is the same as the one absorbed by convection ($Q_{conv} = Q_{cond}$) resulting in this equation:

$$T_{cell,surface} = \frac{k \cdot \overline{T_{cell,internal}} + h \cdot d \cdot T_{amb}}{h \cdot d + k} \tag{13}$$

The battery lifetime model is based on the (Ah)-throughput approach to calculate the expected lifetime of the battery bank. Moreover, an operating degradation factor is here added to consider also the irreversible chemical phenomena that occur in the electrochemistry of the device.

Moreover, an operating degradation factor is here added to consider also the irreversible chemical phenomena that occur in the electrochemistry of the device. A parameter is calculated in function of temperature, SOC and capacity [30]. Here, only temperature dependence is assessed:

$$k_{degradation}(T, SOC, C) = k_{ref} \cdot f(T) \cdot f(SOC) \cdot f(C) \tag{14}$$

where $k_{ref}$ is a reference parameter defined and fitted for reference conditions (25 °C) in [30]. The temperature dependence ($f(T)$) is modelled according to the Arrhenius equation:

$$k_{degradation}(T) = k_{ref} \cdot e^{\left[\frac{\pm Ea}{R_{gas}} \cdot \left(\frac{1}{T} + \frac{1}{T_{ref}}\right)\right]} \tag{15}$$

where *Ea* (J/mol) is the activation energy of the chemical reactions and $R_{gas}$ (J/mol K) is the universal gas constant.

PEMFC system. The power consumption and production of a PEMFC system are evaluated considering three different subsystems, namely the PEM fuel cells (the power source), PEM electrolyser (the hydrogen producer), and the hydrogen tank. The PEMFC and PEM ideal voltages are computed according to the Nernst equation [31]. The hydrogen consumption is finally evaluated.

Shelter thermal model. In order to increase the peculiarity of the work, which is dedicated to the "mission-critical" telecommunication sector, a thermal model of the shelter, typically hosting the telecommunication devices, is implemented in the tool. This is an important improvement, against the traditional and general sizing software available on the market, which permits a better understanding of the behavior of the whole system from a consumption and maintenance point of view. The resulting energy balance is:

$$Q_{devices} - UA_{irradiated}\left(T_{mean} - T_{sol-air}\right) - UA_{shadowed}\left(T_{mean} - T_{ext}\right) - Q_{cooling} + \frac{m_{air}c_p\left(T_{int,i} - T_{int,f}\right)}{\Delta t} = 0$$

where $T_{sol-air}$ is the solar air temperature (the temperature that experiences a surface subject to solar irradiance and convective air flux), evaluated as follows

$$T_{sol-air} = T_{amb} + \frac{\alpha \cdot G \cos \theta_Z - Q_{irr}}{h_{conv}}$$

while $Q_{devices}$ (W) represents the power dissipated by the radio devices, $Q_{cooling}$ (W) is the cooling power, $U$ is the shelter wall transmittance (W/m$^2$ K), $A_{irradiate}$ (m$^2$) and $A_{shadowed}$ (m$^2$) are respectively irradiate and shadowed areas, $T_{mean}$ (K) is the internal mean temperature between the initial and final ones ($T_{int,i}$ and $T_{int,f}$ respectively) and $m_{air}c_p$ (J/K) is the heat capacity of the internal air.

Dispatch strategy. The HPS can operate according to two different dispatch strategies:

a.   Load following (LF): the batteries are only charged whenever the power from RES exceeds the primary load and the DG will furnish the power to meet the load.

b.   Cycle charging (CC): the DG operates at maximum power to supply the load and charge the batteries up to a pre-defined SOC.

The proper dispatch strategy for HPS architecture is a further design variable to be optimized [32].

## 4. Field Test Case

The asset criticality analysis is carried out over 70 real sites which constitute the overall network. The result is a series of listed sites with the associated CI value as output. In this paper only the worst case is considered, named *Test Case Site* with a CI equal to 9.81/10.

### 4.1. Input Parameters

The input parameters used by the algorithm and specific ranges considered in the variable spread are reported in Table 2. The selected boundaries related to the main design variables are reported in Table 3. In particular, with respect to the battery model at that moment, we decided to keep fixed variable x5 using only Lithium-Ion models since they are the most reliable and commonly used in today's industrial market.

**Table 2.** Input parameters and mutation variables.

| | | |
|---|---|---|
| **GA** | Population size | 100 |
| | Number of generations | 50 |
| | Number of runs | 2 |
| | Crossover Index | 10 |
| | Mutation Index | 10 |
| | Mutation probability | 1/NUM [variables] |
| **Ambient quantities** | GHI radiation (W/m$^2$) | Min = 0 Max = 1188 |
| | Ambient temperature (°C) | Min = −20.4 Max = 23.9 |
| | Wind velocity (m/s) | Min = 0 Max = 8.8 |

**Table 3.** Variable constraints adopted in the design optimization process.

| Variables | Thresholds | Units |
|---|---|---|
| $x_1$ | [8; 15] | PV panels |
| $x_2$ | [45; 55] | Degrees |
| $x_3$ | [170; 190] | Degrees |
| $x_4$ | [4; 8] | Batteries |
| $x_5$ | - | Lithium-Ion |
| $x_6$ | [20; 240] | diesel liters |
| $x_7$ | [3000; 9000] | $H_2$ liters |
| $x_8$ | [0; 1] | CC/LF |
| $x_9$ | [0; 1] | Y/N |

Table 4 shows the main characteristics of the components in HPS available on datasheets.

**Table 4.** Technical data of the HPS components.

| | | |
|---|---|---|
| **PV** | Cell technology | Mono-Si |
| | Nominal power | 350 kWp |
| | Temp. coefficient | −0.258 %/°C |
| | STC Temperature | 25 °C |
| | Power reduction coeff. | 0.9 |
| | Lifetime | 25 years |
| | Cost | 280 EUR/panel |
| | O&M costs | 1.5% of capital cost |
| | Indirect $CO_2$ | 0.059 kg $CO_2$-eq/kWh |
| **WT** | Technology | Vertical 3-blades |
| | Rated power | 600 W |
| | Generator type | 3-phase permanent magnet |
| | Material | Aluminium Alloy |
| | Lifetime | 25 years |
| | Cost | 2800 EUR/system |
| | O&M costs | 2% of capital cost |
| | Indirect $CO_2$ | 0.02 kg $CO_2$-eq/kWh |
| **DG** | Technology | IC |
| | Nom. power | 1 kW |
| | Intercept | 0.084 L/h/kW$_{rated}$ |
| | Slope coeff. | 0.246 L/h/kW |
| | Lifetime | 2000 h |
| | Diesel LHV | 43.2 MJ/kg |
| | Cost | 900 EUR |
| | Indirect $CO_2$ | 454.3 kg $CO_2$/kW |
| | Direct $CO_2$ | 2.64 kg $CO_2$-eq/L |

**Table 4.** *Cont.*

| | | |
|---|---|---|
| **PEMFC** | Technology | Proton-exch. membrane |
| | Rated power | 450 W FC/1 kW Elect. |
| | Costs | 4840 EUR FC/9070 EUR Elect. |
| | Tank cap/Tank costs | 3000 litre/4990 EUR/tank |
| | Lifetime | 15,000 h |
| | O&M costs | 1.5% of capital cost |
| **BESS** | Technology | Li-Fe-Po$_4$ |
| | Nom. Cap. | 2.4 kWh |
| | DoD | 90% |
| | Lifetime | 11 years (at 25 °C) |
| | Costs | 1100 EUR/battery |
| | O&M costs | 0.5% of capital cost |
| **PLANT** | Lifetime | 25 years |
| | Assembling cost | 3000 EUR |
| | Ancillary cost | 3% of capital cost |
| | Electric cost | 0.18 EUR/kWh |
| | Extra O&M | 2000–3000 EUR special vehicle 300 EUR off-grid vehicle 0 EUR standard vehicle |

The main parameters related to the economic assessment and NPC computation are the nominal interest rate (4%), inflation rate (2.5%) and actual interest rate (1.46%). The NPC is computed using the formula proposed in Section 3, computing every year the Cash Flow Out and Cash Flow In (see Table 5), actualizing the result with an interest rate and applying a differential cost method, evaluating the cash flow of two different alternatives: respectively adopting (Out) or non-adopting (In) the off-grid HPS. With these hypothesis positive NPC indicates economic non-viable solution, while negative NPC cost-effective proposal, which could bring about a possible money-saving for the company.

**Table 5.** Cash Flow main IN and OUT data.

| **Cash Flow out** | **Cash Flow in** |
|---|---|
| - Initial investment (paid at year 1) - Ordinary O&M - Fuel cost - Battery replacements | - Yearly electric bill - Extraordinary maintenance |

Starting from the TMY, a month related to a specific year that represents the meteorological variables is selected.

*4.2. Test Case Site Optimization Results*

The complete set of combinations are analysed during the sizing and optimization processes; Figure 4 shows the Pareto plot of LPSP obtained without PEMFC.

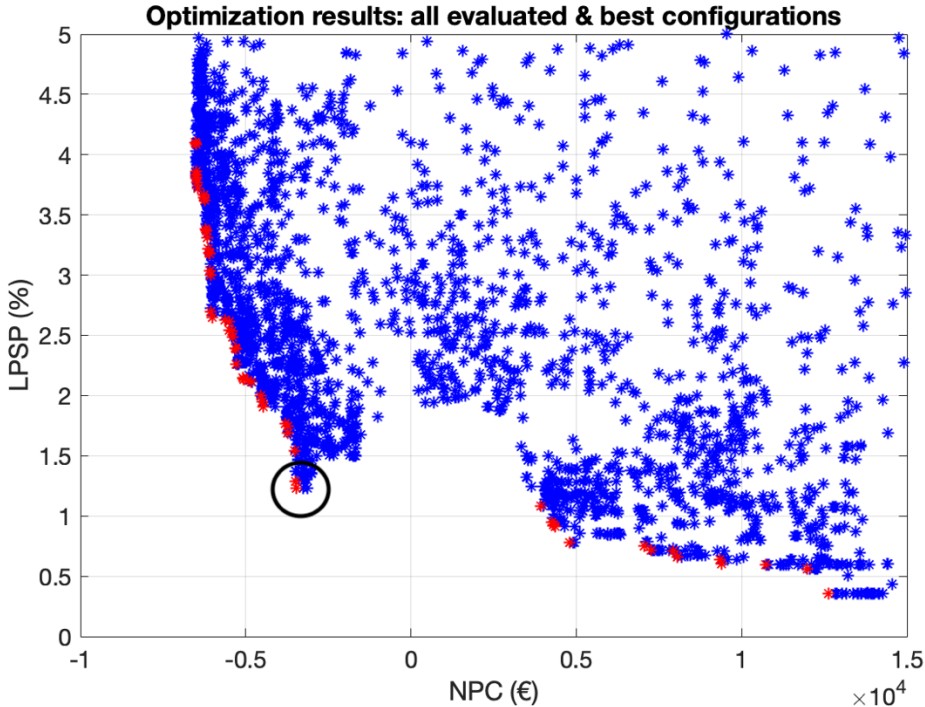

**Figure 4.** Pareto plots of all the examples evaluated (blue dots) and best configurations (red dots). In the black circle is the best solution.

Table 6 reports the site composition used during the sizing process and profile load definition.

**Table 6.** Load site definition considering nominal value and hypothesized duty cycle.

| Constant Consumption Devices | | Intermittent Consumption Device | | |
|---|---|---|---|---|
| Device | Cons. | Device | Cons. | DC |
| Backbone | 160 W | 118 | 150 W TX/20 W SB | 20% |
| Tetra | 100 W | AIB | 150 W TX/20 W SB | 10% |
| Apparatus | 70 W | PCV | 100 W TX/20 W SB | 5% |
| | | Air ext. | 100 W | 5% |

The computational time necessary to complete the simulation of 10,200 possible plant configurations is equal to 2295 s (almost 40 min).

The optimization procedure provides as output a configuration consisting of:

- a PV array composed by 12 panels, with tilt angle of 45° and azimuth of 170°;
- a fixed vertical axis wind turbine;
- five lithium-ion batteries;
- a diesel tank of 73 litres capacity.

The Cycle Charging is the selected energy management strategy without PEMFC. The resulting NPC is equal to −3057 EUR, showing a potential economic saving, while the expected LPSP is 1.13%.

*4.3. System Stress Test*

The HPS implementation is tested, verifying the performance of the system under the critical year of 2018. In particular, at the end of October, extreme wind intensity and extraordinary heavy rain caused damage to the supply electrical systems, resulting in numerous blackout.

A simulation run is performed assuming a starting SOC of 0.5 for the battery pack in the first month of the year and 90% of the diesel tank capacity. The related results are reported as follows.

Figure 5a shows the power produced by the PV plant, in general, it is equal to the expected one, with the exclusion of few periods where the radiation availability is very low compared to the seasonal average values. The total energy produced in 2018 was 5130 kWh. The WT power output (see Figure 5b) is at its baseline due to the low value of average wind speed and accounts for 443 kWh.

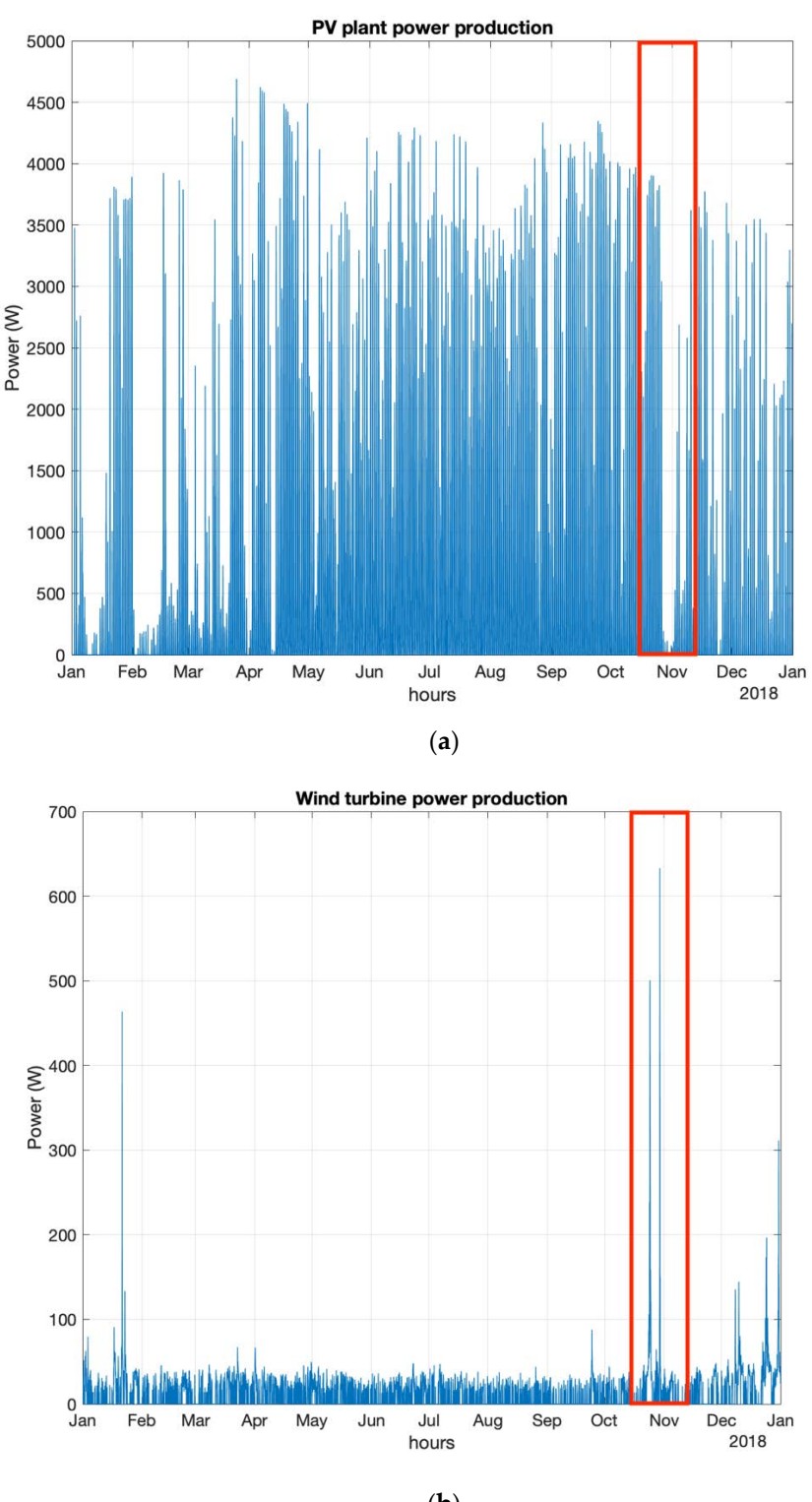

**Figure 5.** (**a**) PhotoVoltaic (PV) and (**b**) wind turbine power production in 2018. Highligted critical period for extreme weather conditions (red box).

The solar radiation strongly influences the battery SOC. Figure 6 shows the internal and battery temperature. The shelter temperature keeps above the external one due to the internal equipment heat dissipation (Figure 7).

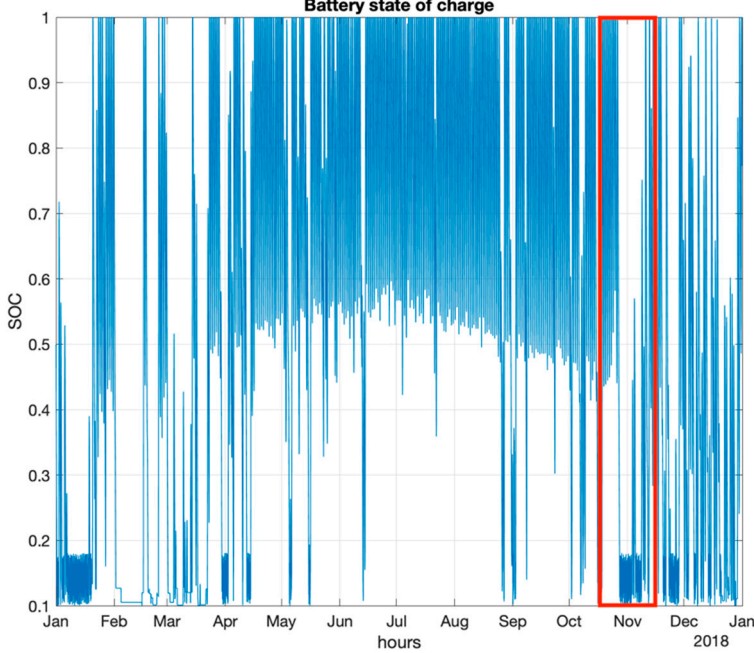

**Figure 6.** Battery State of Charge (SOC) trend in 2018. Highligted critical period for extreme weather conditions (red box).

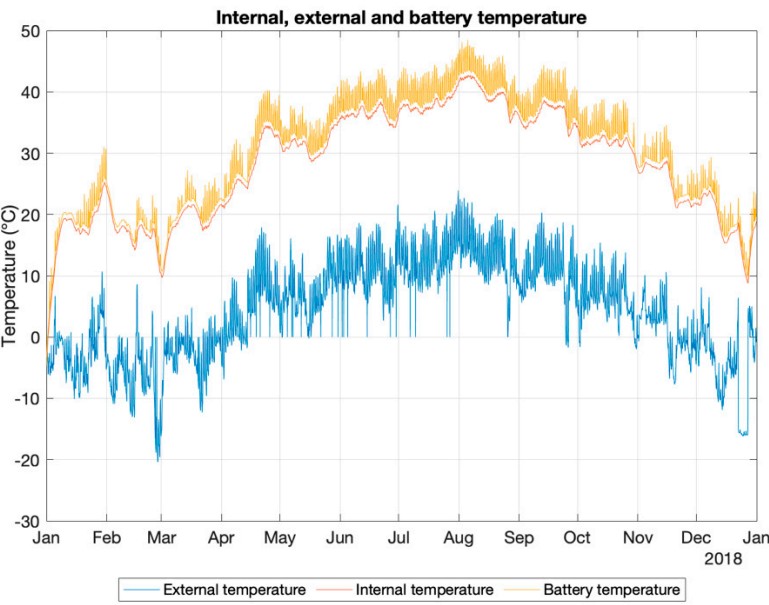

**Figure 7.** Trend of the temperature in the shelter (internal), outside (external) and of the battery in 2018.

Considering the nominal consumption values of each radio apparatus, the heat power lost by the radio systems is 350 W. The internal temperature never exceeds the critical limit of 50 °C for the preservation of electronic devices. The diesel level (Figure 8) follows the same trend: low level during the winter period, due to intensive operation, and high-level during the summer period when PV plant power production is enough to cover the entire load. The LPSP value reaches 10% during this

particular year, resulting in several blackout hours (up to 922 h). Figure 8 shows an intensive usage of the IC engine related to the emergency in November 2018.

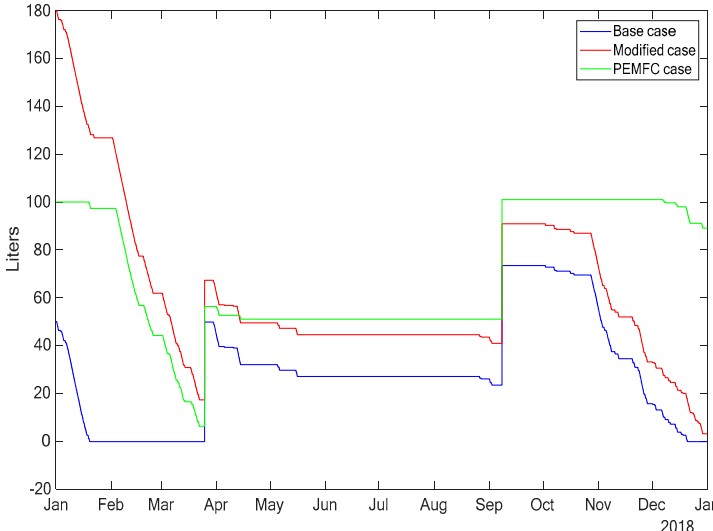

**Figure 8.** Comparison of the level of the diesel fuel in the tank in the three cases: base case, modified capacity case and Proton Exchange Membrane Fuel Cell (PEMFC) case.

### 4.4. System Stress Test with PEMFC

The use of PEMFC system in HPS was analyzed, with a particular focus on the critical month of October. The main benefit of such a system is to save the surplus of energy during the summer period, increasing the redundancy of the system and extending lifetime thanks to the stored energy from RES. On the other side, major drawbacks are still represented by relevant O&M and equipment costs.

Figure 9 reports the trend of the hydrogen tank level of the plant with a PEMFC system in 2018. In this case, the PEM electrolyser charges the tank during the summer period, when mainly the PV systems gives an anergy surplus. PEMFC runs mostly during the winter period (in the emergency period for 2018), improving the energy efficiency of the plant, and nullifying the LPSP index. Thanks to PEMFC, there is also a reduced diesel consumption (see Figure 8) and as a consequence the possibility of reducing the diesel tank capacity.

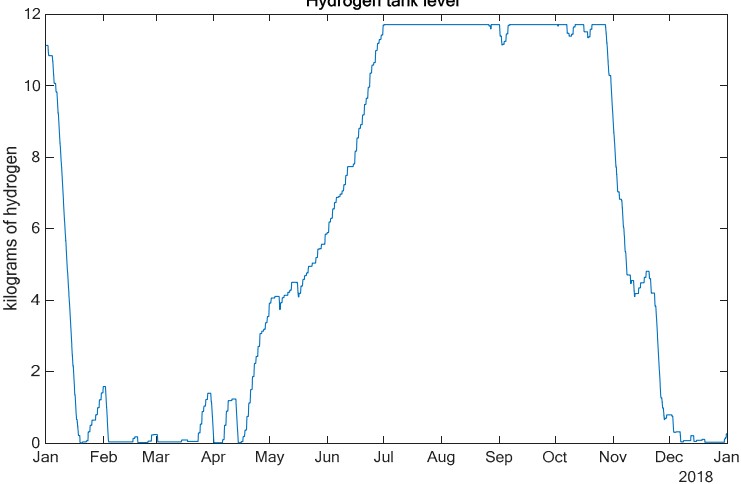

**Figure 9.** Hydrogen tank level in 2018 in HPS system with PEMFC.

Finally, Figure 10 shows a possible site configuration: PV, wind, battery pack, diesel generator and PEMFC are integrated in a unique system.

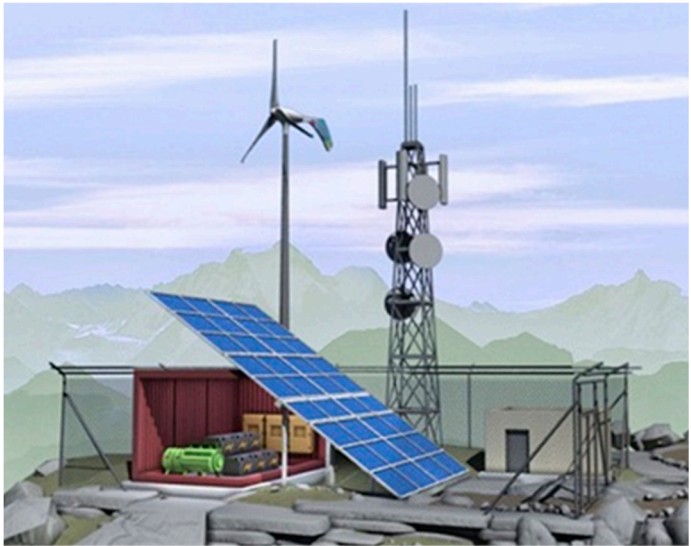

**Figure 10.** Possible site configuration based on HPS with PEMFC for Radio Networks in Mission-Critical Applications.

## 5. Conclusions

This manuscript presents a new procedure to select the most relevant sites in a wide and complex radio network for Mission-Critical (MC) communications, where it is potentially profitable to replace the traditional power supply with new hybrid renewable systems. The choice of these strategic sites is linked to technical and geographical aspects with the aim at the same time to guaranteeing proper continuity of service and reducing operational and maintenance costs. Thus the authors set up an ad hoc optimization tool specifically developed and tested for MC networks.

Furthermore, based on real data, the paper analyzed both the technical and economic feasibility to supply energy to remote sites in an off-grid mode, adopting alternative and renewable power systems with the last technologies available on the market to make the site energetically self-sufficient.

Reported results evidently show a cost-ineffective behavior to adopt PEMFC systems in this type of plant because of the high CAPEX and low cycle energy efficiency of the components. In fact, producing hydrogen from water by means of electrolysers is an energy-intensive and not efficient method. Moreover, the fuel cell shows higher efficiency than the internal combustion engine though it remains around 70%. The use of a PEMFC system generally is not so cost-effective even if the number of extraordinary maintenance interventions with special vehicles (e.g., helicopters) heavily affects this result. However, expensive storage systems and programmable loading together with renewable energy systems can often become profitable for the energy supply of a remote telecommunication site. These systems can in fact reduce diesel run time and therefore increase the impervious site reliability with renewable sources under certain constraints to avoid high Loss of Power Supply Probability (LPSP) values.

**Author Contributions:** Conceptualization, S.L., M.M. and M.R.; methodology, M.M. and M.R.; software, M.R.; validation, M.R. and F.G.; formal analysis, S.L. and M.R.; investigation, M.M. and M.R.; resources, S.L.; data curation, S.L. and F.G.; writing—original draft preparation, S.L. and F.G.; writing—review and editing, S.L. and F.G. All authors have read and agreed to the published version of the manuscript.

**Funding:** This research received no external funding.

**Conflicts of Interest:** The authors declare no conflict of interest.

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
