# Peer review of "Hybrid Power System Optimization in Mission-Critical Communication"

_electronics, doi:10.3390/electronics9111971_

Round 1
Reviewer 1 Report
Hybrid power system optimization is crucial for renewable energy development. The research is interesting. I have some comments as follows:
1. Introduction part should be improved with more discussions of heuristic optimization methods and related renewable energy applications. The difference of your article in comparison with the existing ones should be also highlighted.
2. Some photos are too blurred and should be improved so that the readers can better understand.
3. When describe heuristic optimization methods and related renewable energy applications, please considering these highly-related works: Optimal charging control for lithium-ion battery packs: A distributed average tracking approach; Charging pattern optimization for lithium-ion batteries with an electrothermal-aging model. Lithium-ion battery charging management considering economic costs of electrical energy loss and battery degradation.
4. The measurement noise and shift noise would highly affect your optimization results. Have you considered the effects of these noises?
5. How to handle the hard constraints when you design your optimization method?
6. How to achieve uncertainty quantification as the reliable uncertainty management is crucial for the hybrid power system optimization in mission critical communication?
7. When describe the powerful solutions to achieve uncertainty quantification, please considering these highly related works: A Data-driven Approach with Uncertainty Quantification for Predicting Future Capacities and Remaining Useful Life of Lithium-ion Battery; Modified Gaussian process regression models for cyclic capacity prediction of lithium-ion batteries. Gaussian process regression with automatic relevance determination kernel for calendar aging prediction of lithium-ion batteries.
8. How to guarantee your optimization results would not trap into the local optimum conditions? Why there exists some gap between -0.3 and 0.4 NPC of your Pareto plots in Figure 4? Please carefully clarify them.
Author Response
Dear Editor, Dear Reviewers,
Thank you for allowing a major revision of our manuscript, with an opportunity to address all the reviewers’ comments.
We are grateful to the reviewers committee for pointing out so many constructive comments, which really helped us in clarifying the meaning of our work and improving its overall quality.
We are uploading:
(a) our point-by-point response to the comments (in attachment),
(b) an updated manuscript with yellow highlighting changes,
(c) a clean updated manuscript without highlights (PDF main document).
Best regards,
Sonia Leva et al.

Reviewer 2 Report
Thanks authors for interesting article. The paper is written well and easy to read. The topic of paper will be interesting for readers but I have few questions and comments so before being considering for publication, they need also to address the following problems. I also think that the presented in this review contributions are quite significant, so there are several comments to be addressed:
- I can't distinguish the new findings of this paper and the existing approaches in the literature, so reviewer suggests the authors to clearly mention what is novelty of paper compared with existing approaches.
- There are many limitations of the proposed method. Please present a paragraph with the limitation.
- Page 4 line 140 - no reference number to a literature position (there are only information about error existing)
- Item 22 from the list of literature was mentioned, but there is no reference to that in the text of the article.
- Table 2. How were the mean values presented calculated? Because they do not result from the minimum and maximum values presented there.
- Table 3. What the power factor for PV means? After all, it is direct current!
- Table 3. How the life cycle was defined for BEES? The information about the charging cycles and the permitted temperature should be presented because of battery lifetime is dependent on these parameters.
- Figure 5 should be numbered a) and b) respectively
- Page 13 line 341. In opposite to authors the reviewer believes that a temperature of 50 0C significantly reduces the lifetime of batteries.
- Page 13, line 342. The figure 7 was incorrectly quoted instead of figure 8 that should be presented.
- Page 14 line 365 - no reference number to a literature position (there are only information about error existing).
Author Response
Dear Editor, Dear Reviewers,
Thank you for allowing a major revision of our manuscript, with an opportunity to address all the reviewers’ comments.
We are grateful to the reviewers committee for pointing out so many constructive comments, which really helped us in clarifying the meaning of our work and improving its overall quality.
We are uploading:
(a) our point-by-point response to the comments,
(b) an updated manuscript with yellow highlighting changes (in attach),
(c) a clean updated manuscript without highlights (PDF main document).
Best regards,
Sonia Leva et al.

Reviewer 3 Report
The paper presents a feasibility study for powering off-grid radio sites. It is an interesting analysis with many constraints and criteria taken into account and quite extensive results analysis.
The paper lacks however of a clear contribution: the methodology does not contain any new tool and there is no scenarii or frameworks comparison, to make the framework itself a contribution. It's only contribution seems to be the feasability study.
There is in fact a gap between the abstract's promise "The object of this manuscript is to propose and assess a viable solution to optimize the maintenance operations required to restore the proper functionality in such critical and remote site" and the conclusion "This manuscript analyses the feasibility to power a remote radio site in an off-grid mode." It seems to me that the later is closer to the truth.
The work would stand out much more if different optimisation framework were used to show how the proposed framework provides a better solution, or faster solutions, or solutions that can be adapted in real time in case of unexpected changes, or at least a comparison with how it would be done otherwise. Thus the paper would reach a larger audience which for now might be limited to TLC companies. If the contribution is the feasibility study, then in the result part it would be good to insist much more on comparing the scenarios with and without PEMFC and with and without optimisation, and clarify in the introduction that this is the contribution and how it fits in the literature.
The meaning of "pemfc" and in general of all acronyms in the paper should be given.
Author Response
Dear Editor, Dear Reviewers,
Thank you for allowing a major revision of our manuscript, with an opportunity to address all the reviewers’ comments.
We are grateful to the reviewers committee for pointing out so many constructive comments, which really helped us in clarifying the meaning of our work and improving its overall quality.
We are uploading:
(a) our point-by-point response to the comments (in attach),
(b) an updated manuscript with yellow highlighting changes,
(c) a clean updated manuscript without highlights (PDF main document).
Best regards,
Sonia Leva et al.

Round 2
Reviewer 1 Report
The authors answer my most questions. The descriptions of uncertainty quantification are encouraged to be given as it is crucial for hybrid power system optimization.
Author Response
We are grateful to the reviewer for this valuable comment, which helped us in providing more details on the comprehensive method proposed and the continuously refining process.
The approach we used in this research was inspired to the “worst case analysis” approach, as shown in the main input: the weather condition expected.
We added some sentences in Section 3.2.
Reviewer 2 Report
The authors properly addressed all my comments. The paper can be published.
Author Response
Thank you for helping us to improve our paper.